# Virtual Screening of *Artemisia annua* Phytochemicals as Potential Inhibitors of SARS-CoV-2 Main Protease Enzyme

**DOI:** 10.3390/molecules27228103

**Published:** 2022-11-21

**Authors:** Khalid Miandad, Asad Ullah, Kashif Bashir, Saifullah Khan, Syed Ainul Abideen, Bilal Shaker, Metab Alharbi, Abdulrahman Alshammari, Mahwish Ali, Abdul Haleem, Sajjad Ahmad

**Affiliations:** 1Department of Health and Biological Sciences, Abasyn University, Peshawar 25000, Pakistan; 2Institute of Biotechnology and Microbiology, Bacha Khan University, Charsadda 24461, Pakistan; 3School of Biomedical Engineering, Shanghai Jiao Tong University, Shanghai 200240, China; 4Department of Biomedical Engineering, Chung-Ang University, Seoul 06974, Republic of Korea; 5Department of Pharmacology and Toxicology, College of Pharmacy, King Saud University, P.O. Box 2455, Riyadh 11451, Saudi Arabia; 6Department of Biological Science, National University of Medical Sciences, Rawalpindi 46000, Pakistan; 7Department of Microbiology, Quaid-i-Azam University, Islamabad 45320, Pakistan

**Keywords:** Middle East respiratory syndrome coronavirus, main protease enzyme, molecular docking, molecular dynamic simulation, binding free energy estimation

## Abstract

Severe acute respiratory syndrome coronavirus 2 (SARS-CoV-2) is a human coronaviruses that emerged in China at Wuhan city, Hubei province during December 2019. Subsequently, SARS-CoV-2 has spread worldwide and caused millions of deaths around the globe. Several compounds and vaccines have been proposed to tackle this crisis. Novel recommended in silico approaches have been commonly used to screen for specific SARS-CoV-2 inhibitors of different types. Herein, the phytochemicals of Pakistani medicinal plants (especially *Artemisia annua*) were virtually screened to identify potential inhibitors of the SARS-CoV-2 main protease enzyme. The X-ray crystal structure of the main protease of SARS-CoV-2 with an **N3** inhibitor was obtained from the protein data bank while *A. annua* phytochemicals were retrieved from different drug databases. The docking technique was carried out to assess the binding efficacy of the retrieved phytochemicals; the docking results revealed that several phytochemicals have potential to inhibit the SARS-CoV-2 main protease enzyme. Among the total docked compounds, the **top-10** docked complexes were considered for further study and evaluated for their physiochemical and pharmacokinetic properties. The **top-3** docked complexes with the best binding energies were as follows: the **top-1** docked complex with a −7 kcal/mol binding energy score, the **top-2** docked complex with a −6.9 kcal/mol binding energy score, and the **top-3** docked complex with a −6.8 kcal/mol binding energy score. These complexes were subjected to a molecular dynamic simulation analysis for further validation to check the dynamic behavior of the selected top-complexes. During the whole simulation time, no major changes were observed in the docked complexes, which indicated complex stability. Additionally, the free binding energies for the selected docked complexes were also estimated via the MM-GB/PBSA approach, and the results revealed that the total delta energies of MMGBSA were −24.23 kcal/mol, −26.38 kcal/mol, and −25 kcal/mol for **top-1**, **top-2,** and **top-3**, respectively. MMPBSA calculated the delta total energy as −17.23 kcal/mol (**top-1** complex), −24.75 kcal/mol (**top-2** complex), and −24.86 kcal/mol (**top-3** complex). This study explored in silico screened phytochemicals against the main protease of the SARS-CoV-2 virus; however, the findings require an experimentally based study to further validate the obtained results.

## 1. Introduction

Coronaviruses have been a prominent cause for the spread of deadly pneumonia in humans since the inception of the twentieth century. In 2003, an eruption of the severe acute respiratory syndrome coronavirus (SARS-CoV) was accompanied by a rise in the Middle East respiratory syndrome coronavirus (MERS-CoV), which killed 10% and 35% of infected humans, respectively [1,2]. SARS-CoV and MERS-CoV are zoonotic viruses that infect bats/civets and dromedaries, respectively [1]. In late December 2019, an emergence of an unusual pneumonia disease from an unidentified source appeared in Wuhan, China. The Chinese public health, clinical, and scientific communities acted quickly to achieve the rapid identification of the responsible virus, and conveyed the viral gene sequence to the rest of the globe [2].

Many laboratories recognized a new coronavirus (nCoV) as the causative agent of this unusual pneumonia. Coronaviruses (CoVs) are a distinct class with encapsulated single-stranded RNA [3]. They cause many diseases in humans and animals that affect the liver, respiratory, digestive, and nervous systems. Fever, coughing, sore throat, dyspnea, tiredness, and malaise are common symptoms. The WHO has provisionally named the causative virus as severe acute respiratory syndrome coronavirus 2 (SARS-CoV-2), and the infection it caused was termed coronavirus disease 2019 (COVID-19) [4].

SARS-CoV-2 is a new virus in the CoV family that gives rise to the COVID-19 disease in humans. The CoV contains an RNA genome and belongs to the Coronaviridae family of the order Nidovirales. These are categorized into four genera, α, β, γ, and δ. There are numerous serotypes for each subtype of coronavirus. Pigs, birds, cats, mice, and dogs are among the animals affected by some of these subtypes. Birds are primarily affected by the γ and δ coronaviruses, whereas mammals are primarily affected by the α and β coronaviruses [5]. Humans are affected by seven different coronavirus forms, including two common species of α coronaviruses (HCoV-229E and HCoV-NL63), two β coronaviruses (HCoV-OC43 and CoV-HKU1), and three more pathogenic species of β coronaviruses (SARS-CoV, MERS-CoV, and SARS-CoV-2). SARS-CoV-2 attaches a spike protein (S-protein) to the angiotensin-converting enzyme 2 (ACE-2) receptor on a human cell surface and enters into the human cell. The spikes on the virus’s surface resemble keys. At the surface of our cells, there are particular locks. If the key fits the lock, then the virus invades, hijacks our construction workforce, and duplicates [6]. The search for a COVID-19 treatment is a prominent topic right now. The SARS (severe acute respiratory syndrome)-CoV-2 main protease (Mpro) is one target that has gained a lot of attention from researchers. Several COVID-19 treatment options have been proposed but the search for viable medications to stop the infection is essential. Drugs that target conserved enzymes such as the main protease (Mpro), papain-like protease (PLpro), nonstructural protein 12 (nsp12), and RNA-dependent RNA polymerase (RdRP) could be wide-ranging and efficacious. The protease enzyme is important for viral biology [7]. Consequently, the viral protease is frequently used as a possible drug target. The Mpro is highly comparable to other proteins, so it could be considered an effective and suitable drug target for targeting coronaviruses [8]. This protein is also involved in the regulation of transcription and translation processes; hence, this is another reason for selecting the protein for structure-based drug designing [9]. Mpro is a homodimer protein consisting of two subunits. Each contains three domains, termed domains I, II, and III. Domains I and II range from residue 8 to 101 and 102 to 184, respectively, and are made up of six antiparallel β-barrels and an antiparallel globular cluster of five alpha helices [10]. At the cleft formed by domains I and II, there is a cysteine–histidine dyad, which, together with the *N*-terminal residues, runs from residue 1 to 70 [11]. Mpro has been previously targeted for designing drugs that could tackle new SARS-CoV-2 variants [12].

Furthermore, formulating antiviral medicines that hinder the SARS-CoV-2 Mpro is possible and could prove useful. Plants generate phytochemicals that assist them in fighting diseases caused by fungi, bacteria, and plant viruses, as well as protecting themselves from consumption by insects and other animals [13]. The phytochemicals in plants typically have positive health impacts when consumed [14]. Fruits, vegetables, grains, and other plant foods may contain these bioactive nutritious molecules, which prove beneficial for human health by lowering the chance of major chronic diseases. According to preclinical, clinical, and epidemiological investigations, phytochemicals may be effective for the treatment of a number of diseases due to their antioxidant and anti-inflammatory characteristics [15].

In medical sciences, computer-aided drug-designing methods are acquiring popularity all over the globe as a result of their sophisticated approach and successful methodologies [16]. They provide a working platform for scientists to investigate a variety of biological phenomena, processes, and molecular interactions [17]. These procedures are primarily cost-effective, and they comprise authentic methods that accurately forecast results [18].

Pakistani plant-based phytochemicals have not been much explored against SARS-CoV-2. Therefore, in this study, a virtual screening of phytochemicals from Pakistani plants was carried out to identify potential inhibitors against the SARS-CoV-2 main protease enzyme. For this purpose, phytochemicals from Pakistani plants that possess antiviral activities were searched for in the literature. After that, a receptor target was chosen. In our case, the inhibitor was derived from phytochemicals while the target was the SARS-CoV-2 main protease enzyme. After these two steps, the virtual screening was performed through a molecular dynamic simulation. Through this simulation, the pharmacokinetic properties and toxicity of the phytochemicals as inhibitors were assessed.

Furthermore, it is significant that the computational screening of drug libraries against a given biological macromolecule can ease experimental analyses as well as speed up new drug discovery against SARS-CoV-2. This study will open new avenues for testing the inhibitory potential of phytochemicals from Pakistani plants against SARS-CoV-2. Further, the shortlisted biological potential leads may be used as parent compounds for further derivative investigations.

## 2. Results

### 2.1. Active Site of the SARS-CoV-2 Mpro

The current research study was carried out to virtually screen phytochemicals from Pakistani plants to identify potential inhibitors against the SARS-CoV-2 main protease enzyme. The structure of the main protease consists of three domains, with the first two consisting of antiparallel beta turns and the third consisting of an alpha helix arrangement. Test results for the conserved domain sequences show that the SARS-CoV-2 catalytic residues are found in a cleft between the first two domains, and the **top-3** docked complexes are graphically displayed in Figure 1, Figure 2 and Figure 3. The binding energy values of **top-1**, **top-2**, **top-3,** and the control **N3** molecule were −7.1 kcal/mol, −7 kcal/mol, −7 kcal/mol, and −6.9 kcal/mol, respectively. All three compounds could be seen well docked deep inside the Mpro pocket. The size of the docked molecules was observed to be vital for adjusting themselves in the pocket and forming a wide range of hydrophilic and hydrophobic interactions. From around the pocket, the majority of the residues were seen to be in close contact with the compound atoms. The important active pocket residues involved in interactions with the compounds were His41, Leu141, Gly143, Ser144, Cys145, His163, His164, Glu166, and Gln192. The bond distance of these residues with the compounds ranged from 1.5 to 3 Å. The control **N3** molecule was also seen to form strong hydrogen bonding with Phe140, Gly143, His164, Gly166, Gln189, and Thr190. The Mpro residues that were seen in interactions with the compounds are presented in Table 1. 

### 2.2. Molecular Docking of the Retrieved Phytochemical with SARS-CoV Mpro

A molecular docking analysis of the phytochemicals with the SARS-CoV-2 Mpro was performed in order to identify potential binders against the SARS-CoV-2 main protease enzyme. **Top-10** docked compounds, along with their structure and binding free energies, were obtained from the docking results (Table 2). Among the **top-10** docked complexes, the **top-3** compounds were prioritized for further study based on the lowest binding affinity.

### 2.3. Selection of Top Compounds for Simulation 

Among the **top-10** docked compounds, three top docked complexes (**top-1** (BBB_26580140), **top-2** (BBB_26580153), and **top-3** (BBB_26580155)) were considered for a further molecular dynamic study. The docked complexes were prioritized on the base of lower energy scores, which were obtained from the docking results. 

### 2.4. Physiochemical Properties of the Selected Compounds

The physiochemical properties of the selected compounds were also assessed. The chemical formula, molecular weight, number of aromatic heavy atoms, fraction Csp3, number of rotatable bonds, number of H-bond acceptors, number of H-bond donors, molar refractivity, and TPSA were the physiochemical properties analyzed for each compound. Additionally, lipophilicity, water solubility, and pharmacokinetic analyses of the compounds were performed. Subsequently, in the pharmacokinetic analysis, the GI absorption, BBB permeant, P-gp substrate, drug-likeness, and medicinal chemistry properties of the selected compounds were also checked. All these details can be found in Appendix A. The compounds were found to have suitable physiochemical properties and were classified as drug-like molecules. The compounds were also reported to show good pharmacokinetics by having a high gastrointestinal absorption and a good oral bioavailability. The compounds can be easily synthesized due to a good synthetic accessibility score. Similarly, the compounds had no pan-assay interference (PAINS) chemical moieties, and were therefore selective in their action. 

### 2.5. Molecular Dynamic Simulation Assay for the Selected Compounds 

The top-docked complexes were subjected to a 500 ns MD simulation using the “AMBER20 software” and the results obtained from the simulation trajectories consisted of (i) the root mean square fluctuation (RMSF), (ii) the root mean square deviation (RMSD), and (iii) the radius of gyration (RoG). In molecular dynamic simulations, peaks in the RMSD represent the movement of the ligand within the binding pocket of the receptor, while the RMSF analyzes how much the structure of the protein deviates from a reference place over the whole simulation time. Additionally, the RoG represents how much the docked complex is compact or relaxed throughout the simulation time. In Figure 4A, the RMSD graph of the **top-3** complexes are shown. The **top-1** complex showed stability throughout the simulation with a maximum RMSD of 1.8 Å. A very low deviation in the start was due to several loops present in the protein’s structure. The RMSD of **top-2** showed a continue rise in the peak until 380 ns, and showed stability over the rest of the simulation time with an average deviation of 4.8 Å. Despite these rising peaks, no major changes in the ligand binding pose were observed; however, the loop region in the receptor may have resulted in these peaks. Likewise, the RMSD of the **top-3** complex showed stability after 100 ns until the end of the simulations, with an average RMSD of 2.2 Å. The average RMSF of the **top-3** complexes were noted as 1.1 Å, 1.9 Å, and 2 Å, respectively. Figure 4B shows straight graphs for each complex, showing no major changes in the backbone structure of the receptors. However, the **top-2** complex showed few deviations at 50–70 ns and at 250 ns, which could be due to the contact of the surface loop region with the solvent. Additionally, the radius of gyration (RoG) graph analyzed the competence and relaxation of the complexes. It can be seen that each complex remained compact and no major changes were indicated throughout the simulation time, as shown in Figure 4C. The MDs results verified that all the predicted compounds remained intact with the receptor and could act as modulators; additionally, these results serve as strong evidence that the predicted compounds may have inhibitory effects on the receptor in experiments.

### 2.6. Binding Free Energy Calculation

The binding free energies for the **top-1**, **top-2,** and **top-3** docked complexes were calculated using the MM-GBSA and MM-PBSA modules of AMBER20. The MM-GBSA calculated a total energy of −24.23 kcal/mol for the **top-1** docked complex, −26.38 kcal/mol for the **top-2** complex, and −25 kcal/mol for the **top-3** docked complex. In the MM-PBSA results, the estimated net free energy was −17.23 kcal/mol, −24.75 kcal/mol, and −24.86 kcal/mol for the **top-1** complex, **top-2** complex, and **top-3** complex, respectively (Table 3). The low net binding energy scores predict that the complexes form stable and strong intermolecular interactions.

## 3. Discussion

This study was conducted with the aim of performing a screening of phytochemicals from *A. annua* as potential inhibitors against the SARS COV-2 main protease enzyme (Mpro). Several previous studies have reported the use of different types of compounds for their affinity against SARS-CoV-2 proteins [19]. Several SARS-CoV-2 protein crystal structures are available and might be used as potent drug targets; among them, we used the Mpro. The Mpro of SARS-CoV-2 is a well-known potential drug target for drug designing due to its vital role in the replication and maturation of the virus [20]. Past studies have shown that vasicine, vasicinone, vasicinolone, vasicol, aniflorine, anisotine, vasnetine, and orientin from *Adhatoda vasica* show a significant binding affinity for the Mpro [21]. Other studies have shown GC376 as a broad-spectrum dipeptidyl inhibitor that could inhibit the function of the Mpro [22]. The **N3** crystal structure with the Mpro, evaluated by both experimental and computational studies, has been reported to have a similar sort of results and interact with the same set of Mpro active site residues [21]. Herein, we utilized the docking approach to analyze the binding potency of several phytochemicals from Pakistani medicinal plants, specifically *A. annua*, against the Mpro. The docking results revealed that there are several phytochemicals that have a proper binding ability at the active site of the Mpro and that may block the replication and maturation of the virus. The phytochemicals have strong binding abilities and interact with the enzyme’s active site residues, enriched by both van der Waals and electrostatic interactions. The binding affinity of the **top-3** compounds were considered better and were subjected to a molecular dynamic simulation to investigate their dynamic behavior. The stability of the drug molecule with the pathogen target is also important, so in the molecular dynamic simulation assay, we analyzed the stability of the selected compounds with the targeted protein to determine whether the compounds had binding stability with target proteins or not. 

According to the molecular dynamic simulation assay, we analyzed that the docked complex underwent no major changes throughout the simulation. Some small changes were observed in the RMSD, RMSF, and RoG values, but at the end of simulations, each complex was observed to have good overall conformational stability. The fluctuations in the systems were noted due to the presence of loop regions in the structure. Similarly, the docking and molecular dynamic findings suggested the complexes as potent inhibitors of 3-chymotrypsin and papain-like proteases [23,24,25,26,27]. We further estimated the binding free energies for additional validation of the docking studies, and the results revealed that the docked complexes attained proper stability during the computational experiments. Although the findings are promising, further experimental validation is required to decipher the actual inhibitory potential of the compounds.

## 4. Materials and Methods

The current research study was designed to virtually screen phytochemicals of Pakistani medicinal plants, especially *A. annua*, to identify potential inhibitors against the SARS-CoV-2 main protease enzyme. The step-wise methodology is presented under the following subheadings.

### 4.1. Retrieval of Antiviral Drugs

The study was commenced with the retrieval of antiviral drugs from Selleckchem Inc., the ZINC database, and the drug bank database. *A. annua*-based plant phytochemicals were retrieved using keywords such as “*A. annua*”, “*A. annua* phytochemicals”, “*A. annua* medicinal plant”, and “*A. annua* plant phytochemicals” [28]. The phytochemicals were retrieved in 3D format and minimized using the same methodology described for the Mpro [29]. All the compounds were then transported to the PyRx tool for the molecular docking studies [30]. 

### 4.2. Receptor and Ligand Preparation

Different phytochemicals of *A. annua* were used in the virtual screening approach and were prepared through several steps to be used in the virtual screening process [31]. Herein, before docking, we firstly prepared the ligand for docking purposes. Prior to docking, hydrogen atoms were added, hydrogen bonding was optimized, and all the atomic clashes were removed though structure minimization to achieve better docking results [32]. The X-ray crystal structure of the Mpro of SARS-CoV-2 with the **N3** inhibitor was obtained from the protein data bank (PDB), with a PDB ID of 6LU7. Chain A and chain B were two chains of the enzyme that were retrieved from protein data bank (PDB) [33]. As both chains were homologous in structure, only chain A was prepared with the help of the UCSF Chimera 1.15 tool for an afterward analysis [34]. Energy minimization of the enzyme was performed for 1500 steps and non-relevant ligands such as water molecules were removed [35].

### 4.3. Molecular Docking Study

Molecular docking, simply referred to as a docking study, is an important phenomenon in computer-based drug discovery [36,37]. Herein, a docking pipeline was employed to virtually screen Pakistani plant phytochemicals against the SARS-CoV-2 main protease enzyme to identify potential inhibitors. There are different types of docking software that can be used for docking purposes [36]. The PyRx tool employs autodock vina and can accurately reproduce crystalized complexes [38,39,40]. The docking technique was carried out using the Pakistani-based phytochemicals described above. The docking calculations were considered over ~100 iterations for each drug molecule and docked at the substrate binding site of the Mpro. To validate the docking studies, **N3** was used as a control. 

### 4.4. Physiochemical and Pharmacokinetic Properties of the Selected Compounds

About 90% of compounds do not reach the market because of poor physiochemical properties. Therefore, the physiochemical properties of each compound were calculated using SwissADME (webserver: http://www.swissadme.ch/index.php, accessed on 25 July 2022) [41]. 

### 4.5. Molecular Dynamic Simulation

A molecular dynamic simulation was performed via the AMBER20 software [42]. A molecular dynamic simulation (MDs) is an in silico simulation approach mainly used for the analysis of the dynamic behavior of docked atoms and macromolecules [43,44]. In a molecular dynamic simulation pipeline, a macromolecule is allowed to undergo dynamic behavior for a specific time period and trajectories of the atoms and molecules are determined by solving Newton’s equations of motion [45]. In this study, the dynamic behavior of the drugs was revealed in a 500 ns computer simulation by using the AMBER20 software (Amber, San Francisco, CA, USA). This was performed in order to decode the drug affinity of the ligands for the receptor enzyme versus time. An appropriate number of counter ions were added to the system to make them charge-neutral. The steepest descent step was used to minimize the energy, while a cubic box size of eight angstroms was considered to solvate the complexes. The force field that was considered for the proteins was FF14SB, and GAFF was used for ligands [46]. CCPTRAJ was used for the trajectory analyses [47].

### 4.6. Binding Free Energy Calculations

The binding free energies of the docked molecules were further validated using MMPBSA [48,49]. This was accomplished using the MMPBSA.py module and calculations were performed over 100 frames [50].

## 5. Conclusions

The successful production of novel drug targets generally needs a surfeit of years of study and millions of dollars. Herein, a fast path was adopted to discover novel drug targets against COVID-19. In the study, molecular docking, MD modelling, and binding free energy calculations were carried out for drug target identification. Among all the phytochemicals of *A. annua*, three compounds were screened for having a binding potency with the main protease enzyme of SARS-CoV-2, and for their ability to block the pathogenesis of SARS-CoV-2. The selected compounds were shortlisted on the basis of the lowest binding energy score, with the compounds with lowest net binding energy score having the proper binding efficacy. The stability of a drug molecule with the target is important in order to prevent the pathogenesis of the disease. An MD simulation analysis was performed to investigate the dynamic behavior of the docked molecule. From the findings of the MD simulation analysis, we concluded that the selected compounds had the efficacy of proper binding and stability, which play a vital role in the prevention of disease. Furthermore, the binding free energy calculations also revealed that the **top-3** selected compounds had a better binding potency with the main protease enzyme of the coronavirus and could stop the activity of the target protein, as the target protein is responsible for causing COVID-19 infection. In conclusion, the data provided in the current study could be promising in this regard, but should be subject to appropriate experimental analyses and validation.

## Figures and Tables

**Figure 1 molecules-27-08103-f001:**
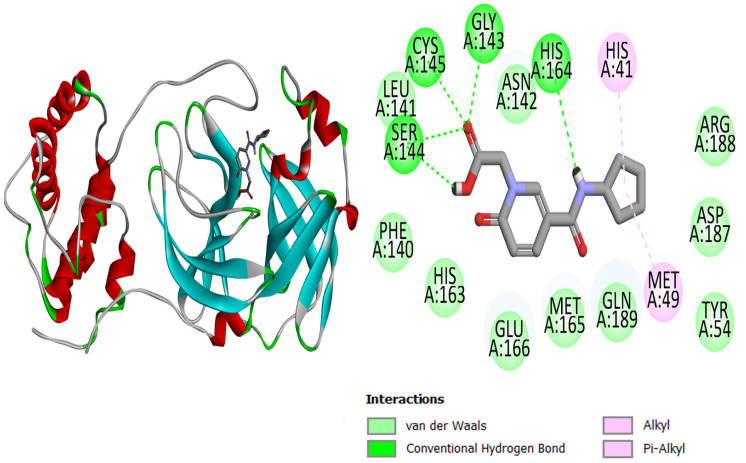
Mpro (shown by cartoon and colored by secondary structure elements) with docked **top-1** compound (shown by stick).

**Figure 2 molecules-27-08103-f002:**
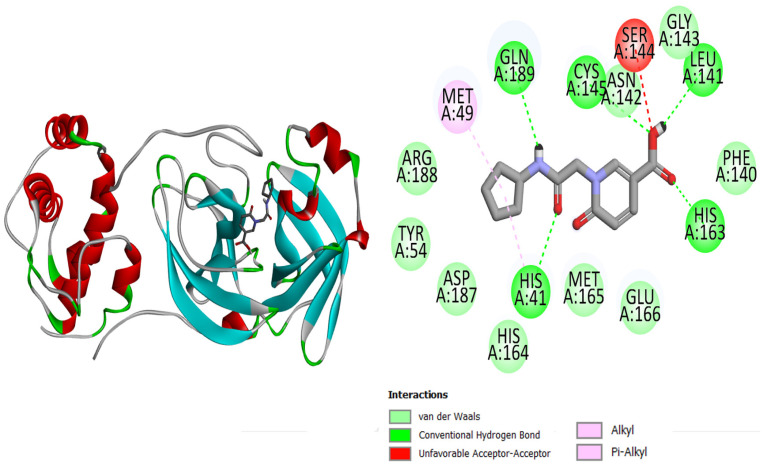
Mpro (shown by cartoon and colored by secondary structure elements) with docked **top-2** compound (shown by stick).

**Figure 3 molecules-27-08103-f003:**
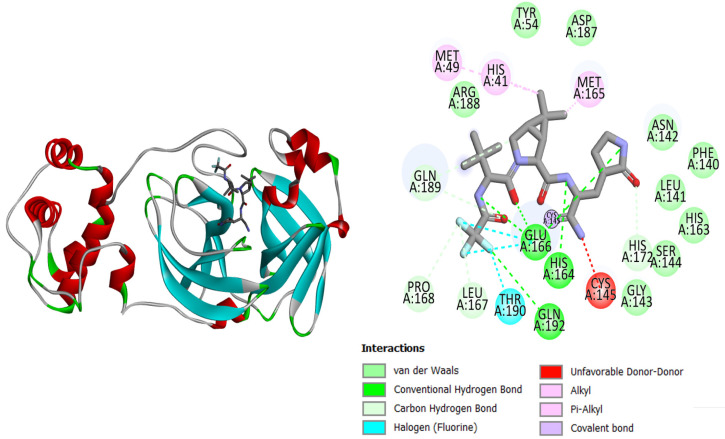
Mpro (shown by cartoon and colored by secondary structure elements) with docked **top-3** compound (shown by stick).

**Figure 4 molecules-27-08103-f004:**
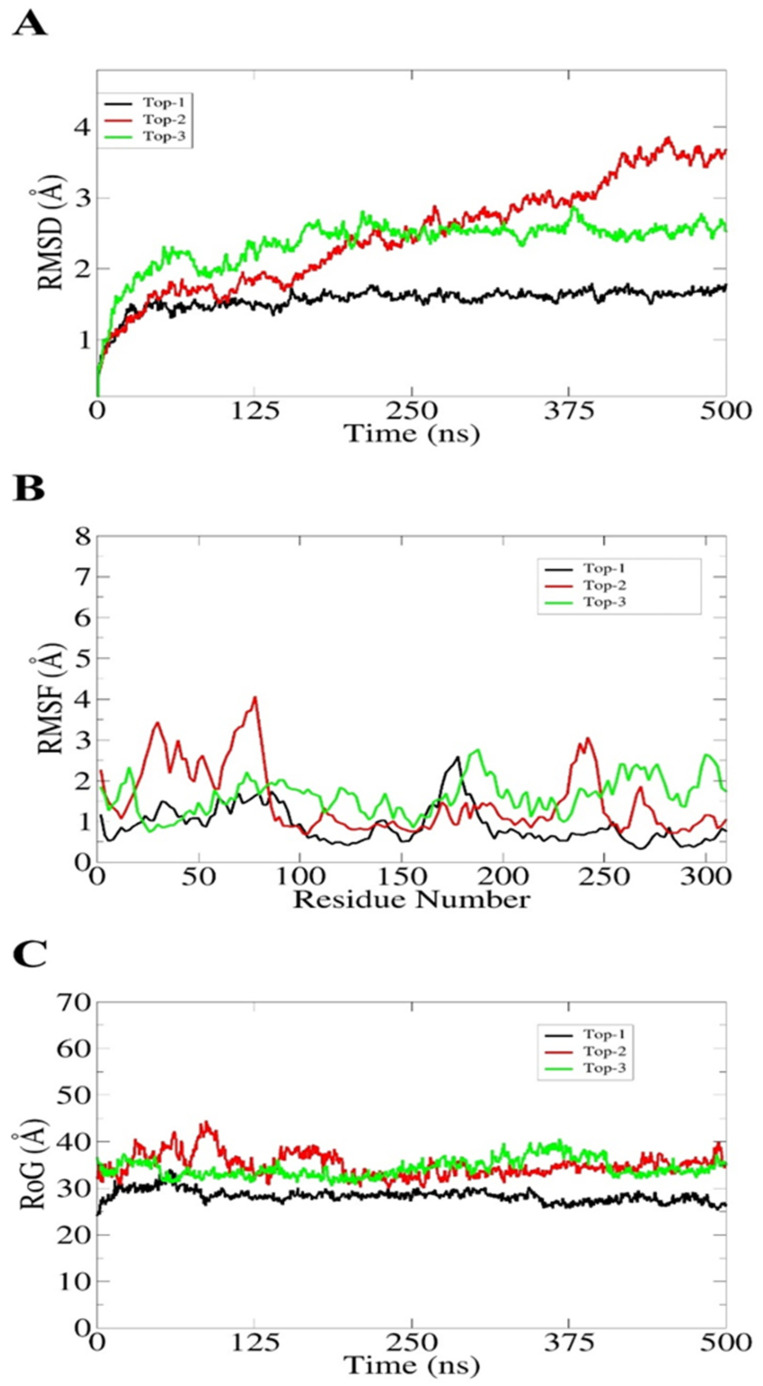
Statistical analysis of simulation trajectories; (**A**) RMSD, (**B**) RMSF, and (**C**) RoG analyses.

**Table 1 molecules-27-08103-t001:** Mpro amino acids seen in interactions with compounds.

S. No	Compound	Interactive Amino Acids
1	**Top-1**	His41, Phe140, Leu141, Asn142, Gly143, Ser144, Cys145, Met149, His163, Met165, Glu166, Asp187, Arg188, Gln189
2	**Top-2**	His41, Tyr54, Leu141, Asn142, Gly143, Ser144, Cys145, His163, His164, Met165, Glu166, Asp187, Arg188
3	**Top-3**	Met49, Tyr54, Phe140, Gly143, Ser144, Cys145, Met165, His164, His172, Asp187, Gln189
4	Control (**N3**)	Leu14, Thr24, Thr26, Tyr54, Phe140, Asn142, Gly143, Ser144, Cys145, His163, His164, Glu166, His172, Gln189, Thr190, Gln192

**Table 2 molecules-27-08103-t002:** **Top-10** docked complexes obtained from docking results; binding energies are shown in kcal/mol.

S. No	Compounds	Structure	Binding Energy Score (kcal/mol)
1	BBB_26580140 (**top-1**)	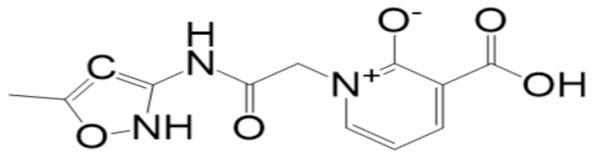	−7.2
2	BBB_26580153 (**top-2**)	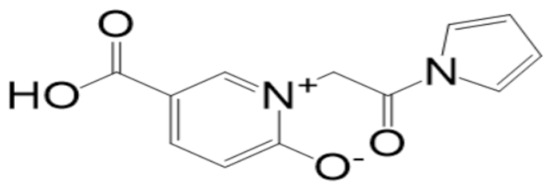	−7
3	BBB_26580155 (**top-3**)	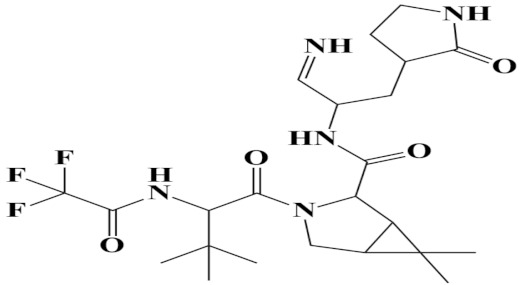	−7
4	BBB_26580162 (**top-4**)	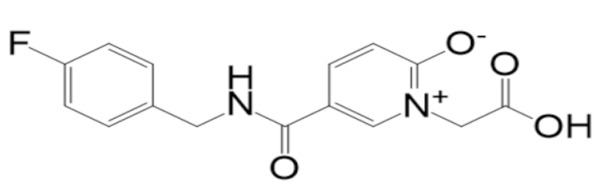	−6.9
5	BBB_265801565 (**top-5**)	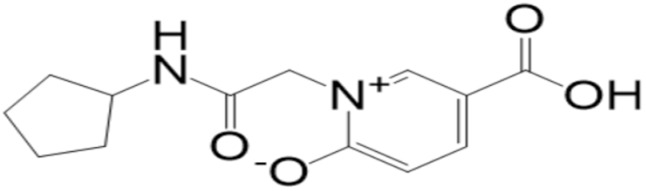	−6.8
6	BBB_26580166 (**top-6**)	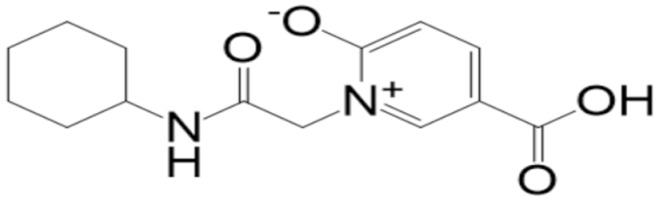	−6.3
7	BBB_26580172 (**top-7**)	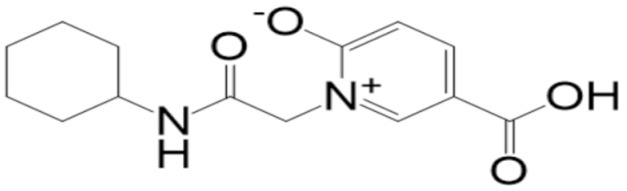	−6
8	BBB_26580189 (**top-8**)	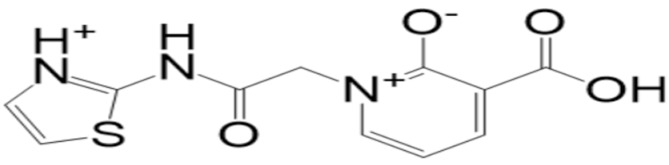	−5.6
9	BBB_26580191 (**top-9**)	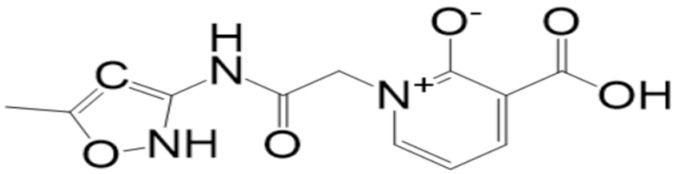	−5.2
10	BBB_29843330 (**top-10**)	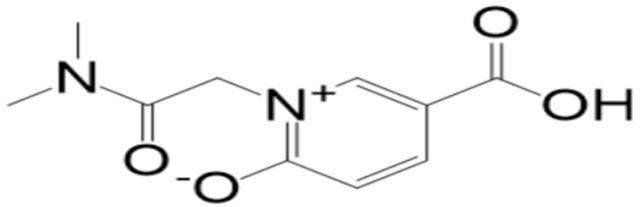	−5.1

**Table 3 molecules-27-08103-t003:** Free binding energy estimation for selected docked complexes.

Energy Parameter	Top-1 Complex	Top-2 Complex	Top-3 Complex
MM-GBSA			
VDWAALS	−27.11	−22.66	−24.69
EEL	−15.90	−15.85	−12.97
EGB	22.77	15.62	18.64
ESURF	−3.99	−3.49	−5.98
Delta G gas	−43.01	−38.51	−37.66
Delta G solv	18.78	12.13	12.66
Delta total	−24.23	−26.38	−25
MM-PBSA			
VDWAALS	−27.11	−22.66	−24.69
EEL	−15.90	−15.85	−12.97
EPB	29.11	18.12	16.44
ENPOLAR	−3.33	−4.36	−3.64
Delta G gas	−43.01	−38.51	−37.66
Delta G solv	25.78	13.76	12.8
Delta total	−17.23	−24.75	−24.86

Key: VDWAALS (van der Waals), EEL (electrostatic), EGB (polar solvation energy of MM-GBSA), ESURF (non-polar solvation energy), delta G gas (net gas phase energy), delta G solv (net solvation energy), delta total (net energy of system).

## Data Availability

All the data generated in the study is reported in the manuscript.

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
