# Peer review of "Virtual Screening of Artemisia annua Phytochemicals as Potential Inhibitors of SARS-CoV-2 Main Protease Enzyme"

_molecules, 2022, doi:10.3390/molecules27228103_

Round 1

Reviewer 1 Report

The rationale for the evaluation of phytochemical as anti-invectives is sound and the methodology employed is appropriate. The main issue with this manuscript is that it reads as an undergraduate project i.e. level of basic detail not relevant and needs to be of a higher/professional standard. Discussion is seriously lacking.

Specific comments:

Abstract - search method for phytochemical does not need to be detailed here - could just shorten to relevant search terms through ZINC database.

Introduction - reads like an undergraduate project. Would be far more relevant to provide a more in depth description of the protein - Figures 1/2/3 repetitive so a clear structure of the protein (ribbons) rather than surface fill to show clear understanding of protein structure and pocket details would be better. (Figs 1/2/3 poor quality so need to ensure any new figures are of good resolution).

Results

2.2 Docking with what software?

Table 1 - problem with structures

Table 1/2 and text confusing. BBB-2658014 Top 3 in table 1 becomes Top 1 in text and table 2, likewise top-6 becomes Top 2 and top-7 becomes Top 3. Need to remove Table 2 and sort out compound naming.

2.4 Where is this discussed with respect to the compounds?

2.5 MD - Do not need to state what MD is. Be concise "protein-ligand complexes were subject to 500 ns MD simulation using ? software/programme) and then provide outputs: RMSD, RMSF and DG (computational binding affinity - currently section 2.6)

Discussion - lack of detailed discussion and analysis of results suggesting the authors do not understand the results.

the statement " ....can inhibit the activity of Mpro of SARS-Cov2...." cannot be used as no biological data/evidence to confirm.

Strong binding ability - need Figures (clear), details, types of binding interactions, how do the different functional groups of the three top compounds interact - similarities/differences/anything to distinguish, relevance to protein function etc...

Need a comparison with described Mpro inhibitors (with confined biological activity) to compare protein-ligand complexes and position of binding and types of binding. This may provide some support that the phytochemical have potential.

Material and Methods

Remove Figure 5

This section should just state Methods and Material and not include background details/discussion - see sections 4.3, 4.4 and 4.5

Conclusions - need to develop discussion/analysis of results and then have another go at conclusions

Please carefully read the manuscript for grammatical errors, spelling and format (should be consistent)

Author Response

Response to reviewer comments

We thank to the all Referee for spending time and interest in our work (vaccines-1950786) and for helpful comments that will greatly improve the manuscript. We have checked all the general and specific comments provided by the Referee and have made all the necessary changes according to his indications. Please refer to yellow highlighted sections in the revised manuscript.

Reviewer 1

Comments and Suggestions for Authors

The rationale for the evaluation of phytochemical as anti-invectives is sound and the methodology employed is appropriate. The main issue with this manuscript is that it reads as an undergraduate project i.e. level of basic detail not relevant and needs to be of a higher/professional standard. Discussion is seriously lacking.

Specific comments:

Abstract - search method for phytochemical does not need to be detailed here - could just shorten to relevant search terms through ZINC database.

Response: Done in the revised manuscript.  

Introduction - reads like an undergraduate project. Would be far more relevant to provide a more in depth description of the protein - Figures 1/2/3 repetitive so a clear structure of the protein (ribbons) rather than surface fill to show clear understanding of protein structure and pocket details would be better. (Figs 1/2/3 poor quality so need to ensure any new figures are of good resolution).

Response: Thank you so valuable suggestion. The figures1/2/3 quality is improved in in revised manuscript. The authors feel that presentation of both surface and ribbon structures are significant and valuable therefore retained as such in the revised manuscript.   

Results

2.2 Docking with what software?

Response: Docking analysis was done through PyRx software. Please refer to section 2.4 of the revised manuscript.

Table 1 - problem with structures

Response: In table 1, all structures are corrected in revised manuscript.

Table 1/2 and text confusing. BBB-2658014 Top 3 in table 1 becomes Top 1 in text and table 2, likewise top-6 becomes Top 2 and top-7 becomes Top 3. Need to remove Table 2 and sort out compound naming.

Response: Corrections are done as per the reviewer suggestion. 

2.4 Where is this discussed with respect to the compounds?

Response: Added to the revised section 2.4.  

2.5 MD - Do not need to state what MD is. Be concise "protein-ligand complexes were subject to 500 ns MD simulation using ? software/programme) and then provide outputs: RMSD, RMSF and DG (computational binding affinity - currently section 2.6)

Response: Thank you so much for valuable suggestion, corrected in revised version of manuscript.

Discussion - lack of detailed discussion and analysis of results suggesting the authors do not understand the results.

Response: Thank you for valuable suggestion. Discussion chapter of the manuscript improved with results of several previous analyses.

the statement " ....can inhibit the activity of Mpro of SARS-Cov2...." cannot be used as no biological data/evidence to confirm.

Response: Thank you for valuable suggestion. The statement is corrected in the revised manuscript 

Strong binding ability - need Figures (clear), details, types of binding interactions, how do the different functional groups of the three top compounds interact - similarities/differences/anything to distinguish, relevance to protein function etc...

Response: Done in the revised manuscript.   

Need a comparison with described Mpro inhibitors (with confined biological activity) to compare protein-ligand complexes and position of binding and types of binding. This may provide some support that the phytochemical have potential.

Response: Done in the revised manuscript discussion section.  

Material and Methods

Remove Figure 5

Response: Thank you for suggestion. Figure 5 is removed from revised manuscript.

This section should just state Methods and Material and not include background details/discussion - see sections 4.3, 4.4 and 4.5

Response: Done in the revised manuscript.  

Conclusions - need to develop discussion/analysis of results and then have another go at conclusions

Response: Thank you for good suggestion. Conclusion part of the paper is revised as per reviewer suggestion. Main results are incorporated with discussion and concluded the main summary. 

Please carefully read the manuscript for grammatical errors, spelling and format (should be consistent)

Response: The manuscript is read thoroughly for grammar and typo errors. 

Reviewer 2 Report

1.      The manuscript deals with COVID19. There are so many articles coming on this since its arrival. The authors need to justify the importance of their research.

2.      Introduction section need to be improved adding some latest concepts. Structural features for Mpro need to be added, the authors can aid with these articles for the same.

https://doi.org/10.1042/BSR20201256

https://doi.org/10.1016/j.ijbiomac.2021.02.071

3.      Erratic abbreviation usage must be avoided. At first places, the authors are strictly advised not to use abbreviations.

4.      Among total docked com-pounds, top-10 docked complexes were considered for further study and evaluated for their physiochemical and pharmacokinetics properties. What was the criteria for this?

5.      Figure 1 and 2 quality needs to be improved.

6.      Mpro was performed in order to identify potential binder against SARS-cov-2 main protease enzyme. cov-2 is written somewhere in uppercase and somewhere in lowercase. The entire manuscript needs to be proofread using software.

7.      Among top 10 docked complexes, top 3 compounds are  prioritized for further study on the base of high binding affinity. What were the criteria used?

8.      Among top- 10 docked compounds top- 3 docked complexes were considered for  further study, the docked complexes were prioritized on the base of lower energies score  which is obtained from docking results is the selected compounds are mentioned in Table  2. Did the authors only considered binding energy?

Author Response

Response to reviewer comments

We thank to the all Referee for spending time and interest in our work (vaccines-1950786) and for helpful comments that will greatly improve the manuscript. We have checked all the general and specific comments provided by the Referee and have made all the necessary changes according to his indications. Please refer to yellow highlighted sections in the revised manuscript.

Reviewer 2

Comments and Suggestions for Authors

  1. The manuscript deals with COVID19. There are so many articles coming on this since its arrival. The authors need to justify the importance of their research.

Response: Justification is added to the revised manuscript introduction section.

  1. Introduction section need to be improved adding some latest concepts. Structural features for Mpro need to be added, the authors can aid with these articles for the same.

https://doi.org/10.1042/BSR20201256

https://doi.org/10.1016/j.ijbiomac.2021.02.071

Response: Added in revised manuscript.

  1. Erratic abbreviation usage must be avoided. At first places, the authors are strictly advised not to use abbreviations.

Response: Corrected as per suggestion.

  1. Among total docked com-pounds, top-10 docked complexes were considered for further study and evaluated for their physiochemical and pharmacokinetics properties. What was the criteria for this?

Response: Dear reviewer, among all docked compounds generated by PyRx tool, we considered Top-hits (Top docked compounds on the base of lowest binding energy score, the compound having lowest negative binding energy score was considered best docked compounds and processed further for molecular dynamic simulation.       

  1. Figure 1 and 2 quality needs to be improved.

Response: Quality of Fig 1 and 2 improved for better visibility in revised manuscript. 

  1. Mpro was performed in order to identify potential binder against SARS-cov-2 main protease enzyme. cov-2 is written somewhere in uppercase and somewhere in lowercase. The entire manuscript needs to be proofread using software.

Response: Thank you for valuable suggestions; The manuscript is thoroughly revised for English language.

  1. Among top 10 docked complexes, top 3 compounds are prioritized for further study on the base of high binding affinity. What were the criteria used?

Response: Thank you. Dear reviewer, the top 3 compounds were prioritized on the base of lowest negative binding energy score, and processed for molecular dynamic simulation analysis. 

  1. Among top- 10 docked compounds top- 3 docked complexes were considered for  further study, the docked complexes were prioritized on the base of lower energies score  which is obtained from docking results is the selected compounds are mentioned in Table  2. Did the authors only considered binding energy?

Response: Yes, there are several others parameters, but we considered binding energy score for best docked solution, the docked compounds have stable binning affinity, having lowest negative binding energy score. Hence we also considered Top- hits on the base of lowest negative binding energy score.

Round 2

Reviewer 1 Report

This revision came back very quickly and is still not up to standard. There is still superfluous details, the Introduction is out of date. Please read some recent literature re. licensed treatment options for COVID-19 and reviews on Mpro inhibitors (you do introduce other Mpro inhibitors in the Discussion, however this should be in the introduction and you do not relate activity with binding interactions etc...) and UG level, the figures are still not clear or very helpful (particularly the colourful blobs), the structures in the Table are inconsistent in format/size. There is still no indication that there is any understanding of the protein itself, for this reason I would strongly recommended you do some reading re. Mpro and this should help you understand the information re binding/interactions, relevance to protein function. Some suggested publications as a starting point:

L. Agost-Beltrán et al. Advances in the Development of SARS-CoV-2 Mpro Inhibitors. Molecules 202227, 2523. 

H.M. Mengist et al. Structural Basis of Potential Inhibitors Targeting SARS-CoV-2 Main Protease. Frontiers in Chemistry 20219, 622898. 

A. Narayanan et al. Identification of SARS-CoV-2 inhibitors targeting Mpro and PLpro using in-cell-protease assay. Communications Biology 2022169, https://doi.org/10.1038/s42003-022-03090-9 | www.nature.com/commsbio 

J. Qiao et alSARS-CoV-2 Mpro inhibitors with antiviral activity in a transgenic mouse model. Science 2021371, 1374-1378.

Jin et al. Structure of Mpro fromSARS-CoV-2 and discovery of its inhibitors. Nature 2020582, 289-293 (and useful supporting information)

Also, for the data to have any relevance, you need comparison with known (potent) Mpro inhibitors (include computational docking/MD/binding affinity for comparison with the phytochemicals described in this paper).

Please take the time to undertake some more studies and achieve a high level publication.

Author Response

Response to reviewer comments

We thank to the all Referee for spending time and interest in our work (vaccines-1950786) and for helpful comments that will greatly improve the manuscript. We have checked all the general and specific comments provided by the Referee and have made all the necessary changes according to his indications. Please refer to green highlighted sections in the revised manuscript.

Reviewer # 1

Comments and Suggestions for Authors

This revision came back very quickly and is still not up to standard. There is still superfluous details, the Introduction is out of date. Please read some recent literature re. licensed treatment options for COVID-19 and reviews on Mpro inhibitors (you do introduce other Mpro inhibitors in the Discussion, however this should be in the introduction and you do not relate activity with binding interactions etc...) and UG level, the figures are still not clear or very helpful (particularly the colourful blobs), the structures in the Table are inconsistent in format/size. There is still no indication that there is any understanding of the protein itself, for this reason I would strongly recommended you do some reading re. Mpro and this should help you understand the information re binding/interactions, relevance to protein function. Some suggested publications as a starting point:

  1. Agost-Beltrán et al. Advances in the Development of SARS-CoV-2 Mpro Inhibitors. Molecules202227, 2523. 

H.M. Mengist et al. Structural Basis of Potential Inhibitors Targeting SARS-CoV-2 Main Protease. Frontiers in Chemistry 20219, 622898. 

  1. Narayanan et al. Identification of SARS-CoV-2 inhibitors targeting Mpro and PLpro using in-cell-protease assay. Communications Biology2022169, https://doi.org/10.1038/s42003-022-03090-9 | www.nature.com/commsbio 
  2. Qiao et al. SARS-CoV-2 Mproinhibitors with antiviral activity in a transgenic mouse model. Science 2021371, 1374-1378.

Jin et al. Structure of Mpro fromSARS-CoV-2 and discovery of its inhibitors. Nature 2020582, 289-293 (and useful supporting information)

Response: All the mentioned papers are cited in the revise manuscript and relevant details are added.

Also, for the data to have any relevance, you need comparison with known (potent) Mpro inhibitors (include computational docking/MD/binding affinity for comparison with the phytochemicals described in this paper).

Response: A control molecule details is added in the revised manuscript docking section. The simulation analysis was not possible due to lack of funding.

Please take the time to undertake some more studies and achieve a high level publication.

Reviewer 2 Report

The manuscript can now be accepted for publication.

Author Response

Reviewer # 2

Comments and Suggestions for Authors

The manuscript can now be accepted for publication.

Response: Thank you for accepting the paper for publication.